Orangutans modify facial displays depending on recipient attention

Waller Bridget M. 1 bridget.waller@port.ac.uk
Caeiro Cátia C. 1 2
Davila-Ross Marina 1
1 Centre for Comparative and Evolutionary Psychology, Department of Psychology, University of Portsmouth , UK
2 Perception, Action and Cognition Research Group, School of Psychology, University of Lincoln , UK
Vonk Jennifer
Electronic publication date: 2015 Mar 19
Publication date: 2015
Volume: 3
Electronic Location ID: e827
Received 2014 Dec 23; Accepted 2015 Feb 17
Copyright: © 2015 Waller et al.
Copyright year: 2015
Copyright holder: Waller et al.
License: This is an open access article distributed under the terms of the Creative Commons Attribution License, which permits unrestricted use, distribution, reproduction and adaptation in any medium and for any purpose provided that it is properly attributed. For attribution, the original author(s), title, publication source (PeerJ) and either DOI or URL of the article must be cited.
License URL: https://creativecommons.org/licenses/by/4.0/

Keywords: Facial expression, Intentionality, Language evolution, Gesture, Primate signals, Emotion, Primate communication, FACS, Primates, Facial displays

Funding: European Commission Leonardo da Vinci Faculdade de Ciências da Universidade de Lisboa and University of Portsmouth University of Portsmouth Department of Psychology Research Committee Forschungszentrum Jülich and Freundeskreis der Tierärztlichen Hochschule Hannover This project was funded by a European Commission Leonardo da Vinci grant (to Cátia Caeiro) for a mobility partnership between Faculdade de Ciências da Universidade de Lisboa and University of Portsmouth, and a research grant from the University of Portsmouth Department of Psychology Research Committee (to Bridget M. Waller and Marina Davila-Ross). The field study was funded by Forschungszentrum Jülich and Freundeskreis der Tierärztlichen Hochschule Hannover. The funders had no role in study design, data collection and analysis, decision to publish, or preparation of the manuscript.

==============================
Primate facial expressions are widely accepted as underpinned by reflexive emotional processes and not under voluntary control. In contrast, other modes of primate communication, especially gestures, are widely accepted as underpinned by intentional, goal-driven cognitive processes. One reason for this distinction is that production of primate gestures is often sensitive to the attentional state of the recipient, a phenomenon used as one of the key behavioural criteria for identifying intentionality in signal production. The reasoning is that modifying/producing a signal when a potential recipient is looking could demonstrate that the sender intends to communicate with them. Here, we show that the production of a primate facial expression can also be sensitive to the attention of the play partner. Using the orangutan (Pongo pygmaeus) Facial Action Coding System (OrangFACS), we demonstrate that facial movements are more intense and more complex when recipient attention is directed towards the sender. Therefore, production of the playface is not an automated response to play (or simply a play behaviour itself) and is instead produced flexibly depending on the context. If sensitivity to attentional stance is a good indicator of intentionality, we must also conclude that the orangutan playface is intentionally produced. However, a number of alternative, lower level interpretations for flexible production of signals in response to the attention of another are discussed. As intentionality is a key feature of human language, claims of intentional communication in related primate species are powerful drivers in language evolution debates, and thus caution in identifying intentionality is important.

Nonhuman primate facial expressions (as well as their human counterparts) have long been considered to be hard-wired, emotional displays (e.g., Darwin, 1872). Facial expressions are often contrasted to nonhuman primate (hereafter primate) gestures, which are not thought to be underpinned by automated, emotional mechanisms, and instead widely believed to be intentional signals (Tomasello, 2008). Such distinctions between the different types of primate communication are used as crucial platforms to develop and support theories of language evolution (see Slocombe, Waller & Liebal, 2011). The behavioural data supporting these conclusions, however, is incomplete in the sense that different modalities are rarely examined within the same methodological framework, and instead each study focuses on a specific modality in isolation (Slocombe, Waller & Liebal, 2011). Moreover, the dichotomy between emotionality and intentionality could be false, and communicative signals might not necessarily be tied to one process or the other (Maiese, 2014; Demuru, Ferrari & Palagi, 2015).

Efforts to understand the evolution of human language, the production of which is highly dependent on intentionality and related abilities such as theory of mind (Hockett, 1960; Dennett, 1989), often search for evidence of intentionality in the communication of other animals in order to identify evolutionary antecedents (Liebal et al., 2004; Leavens, Russell & Hopkins, 2005; Genty et al., 2009; Schel et al., 2013). Definitions of intentionality are debated both within and between groups of philosophers, psychologists, biologists and others, but broadly defined, intentionality refers to acts and thoughts that are goal-directed, voluntary and purposeful (Grice, 1957; Dennett, 1983). The conservative position is that animal communication is not intentional, unless systematic evidence suggests otherwise.

Despite the obvious difficulty in determining when any animal behaviour is intended or not, researchers have attempted to use various observable behaviours to distinguish intentional communication from unintentional, automated communication. Leavens and colleagues (2005) proposed a specific set of criteria, based on those used to categorise the pre-linguistic communication of children: social use (production of the signal is sensitive to a social audience), gaze alternation (the sender looks between the recipient and event/object), attention-getting behaviours (sender attempts to attract recipient’s attention), persistence (repeated use of signal), elaboration (increased intensity and modification of signal to engage recipient) and sensitivity to attentional state (sender adjusts signals depending on visual attention of recipient). These criteria for intentionality have become established in the field of primate communication, but their application varies between studies. For example, some studies use two or more of these criteria as sufficient evidence of intentionality (e.g., Schel et al., 2013), whereas others use only one (e.g., Tempelmann & Liebal, 2012). Some authors argue that multiple (if not all) criteria should be met (Genty et al., 2009; Schel et al., 2013). Attentional stance sensitivity and social use were the most commonly used criteria for identification of intentionality in a recent review of 24 published studies (18 of 24 studies used social use, and 16 of 24 used attentional stance: Liebal et al., 2014). The review found that two studies used attentional stance sensitivity (Hobaiter & Byrne, 2011a; Hobaiter & Byrne, 2011b) and five used social use (Tomasello et al., 1985; Liebal et al., 2004; Hobaiter & Byrne, 2011a; Hobaiter & Byrne, 2011b; Tempelmann & Liebal, 2012) as sufficient criteria for the identification of intentionality in primate’s communication (as in, when only one of several criteria was necessary for intentionality to be attributed).

Social use, therefore, is by far the most commonly applied behavioural criterion for intentionality. Social use, however, has been highly criticised as a true marker of intentional production of a signal, as one could argue that a social audience can trigger communicative behaviours through several low-level mechanisms such as increased arousal, reflexive responses to the presence of another and so on (Liebal et al., 2014). Sensitivity to attentional stance has been argued to be a more resilient marker of intentional communication (Liebal et al., 2014) as communicating more/only when the recipient is capable of receiving the message (i.e., visually attending) could be evidence that the sender has a goal-directed intention to communicate, and even an understanding of the visual perspective of others. However, alternative explanations such as sensitivity to the face and eyes of others, or learning that responses are only achieved when others are facing, are also highly possible. Indeed, the senses and responses of receivers have long been known as important factors in shaping and constraining the evolution and development of communicative signals (e.g., receiver psychology: Guilford & Dawkins, 1991).

Despite the widespread claims that primate facial expressions are used less intentionally than primate gestures, several studies have found that facial expressions can be sensitive to the attentional stance of others. As the primary focus of these studies was often gesture, however, the facial expressions were broader movements sometimes referred to as facial gestures (e.g., head bob, head shake: Liebal, Call & Tomasello, 2004) rather than prototypical facial expressions involving facial muscles. Some studies have also examined responses to a human demonstrator rather than a conspecific in spontaneous social interaction (Poss et al., 2006; Leavens, Russell & Hopkins, 2010). Liebal et al. (2004), however, included prototypical facial expressions in their study of siamang social communication, and found that the vast majority of grins and mouth-opens were produced only in the presence of an attending recipient. L Scheider, BM Waller, L Ona, AM Burrows & K Liebal (2014, unpublished data) have also found that hylobatid facial expressions are longer when facing a conspecific in a variety of social interactions, and Demuru, Ferrari & Palagi (2015) found that bonobos produce play facial expressions more often when their play partner can see them. Moreover, sensitivity to attentional stance in facial signalling may not be restricted to the primate order, as Horowitz (2009) found some evidence that domestic dogs produce facial play signals more often when in the presence of an attentive play partner than an inattentive partner. Demuru, Ferrari & Palagi (2015) argue that such data demonstrate the combined emotional and intentional properties of play facial expressions, building on the neuro-anatomical thesis that emotional and intentional systems both underpin facial expressions (Sherwood et al., 2004) and are intimately intertwined (Cattaneo & Pavesi, 2014).

The focus of the current study, therefore, was to test whether one of the most commonly applied criteria (sensitivity to attentional stance) claimed to demonstrate intentionality can be applied rigorously to the production of a prototypical orangutan facial expression. The relaxed open mouth display (playface, Fig. 1) is a facial expression ubiquitous in the primate order and almost exclusively restricted to play contexts (Van Hooff, 1973; Parr, Cohen & de Waal, 2005). Similar facial expressions are also found in other mammals (see Waller & Micheletta, 2013) and so it seems highly preserved from a phylogenetic perspective. The prototypical form of the playface is similar across primate species, involving an open mouth and exposure of the lower and (in some species) upper teeth (Preuschoft, 1992; Palagi, Antonacci & Cordoni, 2007; Davila Ross, Menzler & Zimmermann, 2008; Palagi, 2008; Waller & Cherry, 2012). The playface has been proposed as a homologue of the human laughter display (Van Hooff, 1972) and as a ritualised form of mock biting during play, as if to demonstrate that play is only play (Bolwig, 1964; Pellis & Pellis, 1996).

Figure 1 Image of orangutan play face.

Example of open mouth facial expression (playface) from supplemental files of Davila Ross, Menzler & Zimmermann (2008).

The characteristics of the playface therefore, are not suggestive of complex underlying cognition, such as developmental sensitivity, flexibility of use, referentiality or intentionality (for a review of the relevance of these features see Liebal et al., 2014). However, even if there is a relatively fixed component to facial expression production, it could quite conceivably still be underpinned by both emotional and intentional processes. For example, catarrhine primates have control of the facial muscles through direct cortical connections, suggesting an element of voluntary control (Sherwood, 2005). Thus, it is possible that primates can modify the playface in response to their audience.

The goal of this study was to use OrangFACS (Caeiro et al., 2013) to determine whether orangutans (Pongo pygmaeus) modify their facial signals depending on the attentional stance of another individual (the recipient) during spontaneous play interactions, and thus meet one of the established criteria for intentionality. We also recorded additional variables to control for potential confounds of more intense play when face-to-face, and responses to the recipient’s facial expression. We recorded playfaces during social play, and examined the influence of (1) the recipients orientation towards the sender, (2) the recipient’s facial expression, and (3) intensity of play, on the complexity (number of facial muscle components) and intensity (extent of mouth opening) of the playface.

Methods

Study area and subjects

The spontaneous dyadic play of 20 orangutans was observed in total. Nineteen individuals (seven females, twelve males; 3–12 years old) featured as focal individuals in the analysis (one was included in the analysis as a recipient only as the roles of focal and recipient were randomly assigned when each play bout was coded, and he never produced facial expressions when allocated the focal role). Twelve individuals were housed at the nursery of the Sepilok Orangutan Rehabilitation Centre (SORC), Malaysia. Inside enclosures consisted of cages where the orangutans stayed overnight either individually or in pairs, and in larger groups during the day. They were taken outdoors (outside their cages) for several hours in the morning and afternoon as part of their training programme where they were filmed during spontaneous play. The remaining eight individuals were semi-free ranging as they had been previously released by SORC into the Kabili Sepilok Forest Reserve. They lived in this forest area during day and night, and were filmed during spontaneous play. Feeding took place three times per day. The nursery-housed orangutans were fed inside their cages. The released orangutans obtained the food from feeding platforms in the forest (provided by SORC), but they were also showing natural foraging behaviours. There was no interaction between the individuals from SORC and the Forest Reserve during the data collection period.

Video data collection

A total of 12 h of spontaneous play behavior was extracted from 39 h of ad libitum (Altmann, 1974) social interaction footage (mean duration = 37 min ± 20.38 SD per individual). Recordings were obtained outdoors between 8 am and 12 pm and between 2 pm and 5 pm from August to October in 2005. Recordings were taken from no more than 10 m away from the play dyad by a handheld video camera, with both animals kept in view as much as possible. Play was identified based on specific play actions (e.g., wrestling, hitting, grappling: Davila Ross, Menzler & Zimmermann, 2008) and only dyadic play that occurred during the footage was extracted (solitary or triadic play was ignored to allow analyses to control for identity of senders and receivers) for the purpose of this study. Research permission was provided by Sabah Wildlife Department and Economic Planning Unit, Malaysia.

Behavioural coding

The video footage was then coded frame-by-frame (25 FPS) using Adobe Premiere Pro CS4 v.4 and Mangold Interact software. In each play dyad, one of the individuals was randomly chosen as the focal individual. All open mouth facial expressions (OMF) were identified using a broad, inclusive operational definition based on OrangFACS (all occurrences where the mouth was opened by AU26 (jaw drop) or AU27 (mouth stretch) to avoid a priori assumptions about the form of play facial expressions). Any OMFs with poor visibility, where the onset was not visible, or where there was physical biting were discarded. OMFs were treated as separate events if the mouth was fully closed for at least 2 s between movements.

The following binomial factors were coded for every OMF, and when any behaviour was not clearly discernable it was marked as unscorable:

(1) Facial orientation: whether the individuals were facing each other and had an unobstructed view of each other’s face within an angle of 45 degrees of head rotation (Fig. 2: face to face, FTF; or not face to face, Not FTF). Each play session was split into multiple periods of FTF and Not FTF play (so each OMF could be classed as FTF or not).

(2) Recipient facial expression: whether the recipient individual displayed an OMF at any point during the duration of the focal OMF (OMFR or nOMFR).

(3) Play intensity: the speed, strength and degree of physical contact of play behaviour between focal and recipient individuals during an OMF (low or high). Play bouts including resting, temporary breaks from play and slow grappling were classed as low intensity play. Play bouts containing chasing, gnawing, grappling, hitting and wrestling were classed as high intensity.

Figure 2 Examples of face to face (FTF) and not face to face (Not FTF) conditions.

To be classed as FTF the two individuals had to have an unobstructed view of each other’s face within an angle of 45 degrees of head rotation.

Facial movement coding

FACS (Facial Action Coding Systems) are useful tools to quantify subtle changes in primate facial signals. The first FACS was developed as an anatomically based observational tool for the measurement of facial movement in humans (Ekman & Friesen, 1978), and has since been modified for use with other animals: chimpanzees (Vick et al., 2007), rhesus macaques (Parr et al., 2010), gibbons (Waller et al., 2012), orangutans (Caeiro et al., 2013) and domestic dogs (Waller et al., 2013). Individual facial muscle movements can be identified and quantified as Action Units (AUs), which allows an objective assessment of morphological changes in facial expressions without the need for a priori emotional labels (Waller & Smith Pasqualini, 2013). Here, OrangFACS (Orangutan Facial Action Coding System: Caeiro et al., 2013) was used to identify the facial movements produced during each OMF. A certified coder (CC) coded whether any of the following action units were present during the OMF using one/zero sampling (Altmann, 1974): brow lowerer (AU4), cheek raiser (AU6), upper lip raiser (AU10), lip corner puller (AU12), lower lip depressor (AU16), jaw drop (AU26) or mouth stretch (AU27). These AUs were chosen as the full range of facial movements likely in an OMF, and all can be present simultaneously in the face with the exception of AU26 and AU27 (which are mutually exclusive).

Reliability coding

To test for data coding consistency, 30 clips of play were extracted at random from the footage and coded (blindly to the study goal) at both the beginning and end of the project (one year apart). Intra-reliability was measured using Cohen’s Kappa (Cohen, 1960). Good agreement was reached for FTF vs. Not FTF (K = 0.66, P < 0.0005) and non-focal OMF (K = 0.77, P < 0.0005). Wexler’s index (Ekman, Friesen & Hager, 2002) was used for AUs, and led to a value of 0.87, which is considered excellent agreement. The inter-observer reliability for play intensity had been assessed previously (K = 0.84, P < 0.0005), and can be considered very good agreement.

Statistical analysis

Generalized linear mixed models (GLMM) were used to analyse our nested data, with defined linear hierarchical groups. The GLMM analysis allowed us to include random factors to control for the fact that: (1) the data set contained missing values (for some observations we could not code for all the factors), (2) individuals appeared a different number of times as both focals and non-focals, (3) more than one OMF was collected from each play bout and/or from the same individual, and (4) not all individuals played together (since the play was spontaneous). We controlled for multiple observations of the same individuals from the same group by adding the identity of the individuals involved in the interaction nested within groups as a random factor and also added a third random factor to control for repeated dyad composition, thus avoiding pseudoreplication (Machlis, Dodd & Fentress, 1985; Pinheiro & Bates, 2009; Waller et al., 2013). The function vif.mer (Frank, 2011) was used to calculate collinearity between the factors. To compute the models, the glmer function from the lme4 package was used (Bates, Maechler & Bolker, 2013). The GLMM were fit by maximum likelihood (ML) with Laplace approximation. Instead of testing for the null-hypothesis to choose our factors, we used an information-theoretic approach (Burnham & Anderson, 2002). We computed ANOVAs and used a combined backward and forward stepwise method, based on Akaike Information Criterion (AIC) to compare models and choose the best one (i.e., the model with the lowest AIC value: Burnham & Anderson, 2002; Field, Miles & Field, 2012). Significance of factors within models was assessed using p-values, which explains the impact of factors on the outcome variable as compared to each other. All the statistical analyses were computed in R 3.1.1 (R Core Team, 2013).

Results

A total of 247 OMFs (see Fig. 1 for an example) were analysed from 19 of the 20 individuals in our sample (mean OMF number ± SD: 13 ± 5.97 per individual, see Table 1), during 121 play bouts (mean bout duration ± SD: 309.48 s ± 482.43). The remaining individual only ever featured as a recipient and so was not included as a focal subject in the analysis. OMF duration ranged from 0.08 s and 10.56 s overall, and there was a significant difference in the duration of OMF in FTF (mean duration ± SD: 1.64 s ± 1.19) and Not FTF (mean duration ± SD: 1.09 s ± 0.71) conditions (Wilcoxon signed ranks test: T = − 2.20, N = 19, P < 0.05). Note that more OMFs were observed than coded (Table 1), as visibility was not always good enough for FACS coding (GLMM analyses are robust and suitable for datasets with missing data).

Table 1 Table of OMF data per individual.

Distribution of OMF events per focal individual in each factor: Face to face, recipient OMF and play intensity. OMFs were discarded whenever FTF or not FTF could not be scored, while unscorable OMFs in recipient OMF and play intensity were maintained and coded as missing data points.

Focal individual	Group (N = nursery,
SF = semi-free ranging)	Age (years)	Face to face	Recipient OMF	Play intensity	
			Yes	No	Yes	No	Low	High	
Anekara	N	2	6	8	3	4	7	7	
Ankong	SF	4	9	8	5	3	1	12	
Annelisa	N	5	4	3	0	2	5	2	
Anpal	N	3	13	4	4	5	10	7	
Boy	SF	7	8	4	4	4	5	7	
Brock	N	3	5	2	3	1	3	4	
Dogi	SF	7	14	3	7	0	5	10	
Kam Chong	SF	8	7	11	3	5	0	17	
Kimbol	N	3	10	8	5	5	7	11	
Mico	SF	5	23	5	12	8	12	12	
Miskam	SF	12	13	4	8	3	5	12	
Naru	N	2	9	7	4	2	6	10	
Nonong	N	3	3	2	1	1	1	4	
Oscar	SF	6	7	4	7	0	0	9	
Patrik	SF	9	2	2	2	0	1	3	
Rosalinda	N	2	3	7	2	2	3	7	
Suzanna	N	3	3	5	4	1	1	7	
Tobby	N	3	10	4	10	2	5	9	
Tompong	N	5	4	3	2	2	4	3	
Totals			153	94	86	50	81	153	
			247	136	234	
Unscorables			(OMFs discarded)	111	13	

As the goal was to compare features of the OMF produced during FTF and Not FTF play, we controlled for the fact that OMF durations differed between conditions (and so the number of AUs produced could differ as a function of time rather than condition if they accumulate over time). We compared the onset latencies of AUs in FTF and Not FTF play in a random subset sample using Wilcoxon matched-pairs signed-rank test. We found no significant difference between the mean latency of AUs onset in FTF (mean latency ± SD: 0.12 s ± 0.13) and Not FTF (mean latency ± SD: 0.14 s ± 0.21) conditions (Wilcoxon signed ranks test: T = 0.00, N = 6, P = 1.000). Therefore, as the start time of all AUs within each OMF is approximately the same time in both conditions (almost immediately at the onset of the expression), the length of the OMF cannot be a factor influencing the number of cumulated AUs, and AUs do not accumulate over time.

Complexity of OMF as a function of recipient attention

To investigate whether complexity of OMF (defined as the total number of individual AUs) varied depending on facial orientation, recipient facial expression and play intensity, we calculated GLMM with Poisson error distribution and log function. The total number of AUs in each OMF was used as a response factor and the identity of focal and recipient individuals as well as the play bout number were entered as random factors. Facial orientation (FTF vs Not FTF), recipient facial expression (OMFR vs nOMFR) and play intensity (low vs high) were entered as fixed factors. There was no overdispersion in the data set or collinearity in the factors.

The model that best fit the data was the full model, containing facial orientation, recipient facial expression and play intensity (see Table 2). The full model was compared to the null model, showing a highly significant difference: ANOVA: F3 = 417.26, P < 0.001. Removal of any of the three factors from the full model resulted in a significant change in the model’s AIC, since all the factors were strongly influencing the facial movement complexity of the focal individual during play behaviour (best model AIC: 454.4 vs model without facial orientation AIC: 459.57, ANOVA: F1 = 7.184, P < 0.001, vs model without recipient facial expression AIC: 812.6, ANOVA: F1 = 360.25, P < 0.001 and vs model without play intensity AIC: 476.15, ANOVA: F1 = 23.767, P < 0.001, Fig. 3).

Figure 3 Figure showing number of OMF in different conditions.

The number of OMF (open mouth facial expressions) containing different numbers of AUs (action units) as (A) a function of facial orientation (FTF, face to face; Not FTF, not face to face), (B) as a function of the facial expression of the recipient (With OMF, recipient has OMF; no OMF, recipient does not have OMF), and (C) as a function of play intensity (low and high).

Table 2 Table of GLMM results.

Optimal GLMM models for the effect of the factors facial orientation, recipient facial expression and play intensity on the facial movement composition of OMF.

Predictor factors	Estimate	SE	z	p	
Response factor: Number of AUs	
Intercept	0.68	0.15	4.55	0.000	
Facial orientation (FTF)	0.36	0.14	2.59	0.009	
Recipient facial expression (OMFR)	0.21	0.11	1.87	0.061	
Play intensity (low)	0.04	0.11	0.33	0.742	
Response factor: AU26 or AU27	
Intercept	0.095	0.51	0.19	0.852	
Facial orientation (FTF)	1.65	0.69	2.38	0.017	
Recipient facial expression (OMFR)	0.18	0.48	0.38	0.707	
Play intensity (low)	−0.61	0.50	−1.23	0.219	

In the full model, facial orientation had a significant positive effect as a fixed factor (P < 0.01): OMF produced when the recipient was facing the sender contained a greater number of AUs (mean number of AUs ± SD in FTF: 3.29 ± 0.11; mean number of AUs ± SD in Not FTF: 2.54 ± 0.11). Facial expression of the recipient was not significant (P = 0.06), but as the AIC was lowered significantly when this was taken out of the model, it had a weak positive effect on the model: OMF produced when the recipient also produced an OMF contained a greater number of AUs (mean number of AUs ± SD in OMFR: 3.35 ± 0.14; mean number of AUs ± SD in nOMFR: 2.60 ± 0.17). Play intensity was also not significant as a fixed factor (P = 0.74). OMF produced when the play intensity was high or low contained a similar number of AUs (mean number of AUs ± SD in high play intensity: 2.97 ± 1.19; mean number of AUs ± SD in low play intensity: 3.04 ± 1.42), but did significantly improve the model fit (although in a positive direction, so low intensity play increased the likelihood of more AUs in the OMF). Each of the factors alone represented a significantly better fit when compared to the null model (facial orientation vs null model ANOVA: F1 = 10.969, P < 0.001; recipient facial expression vs null model ANOVA: F1 = 385.49, P < 0.001; play intensity vs null model ANOVA: F1 = 46.908, P < 0.001) and thus had some impact on complexity of OMFs.

Intensity of OMF as a function of recipient attention

To test whether intensity of OMF varied (whether it contained AU26, a jaw drop, or AU27, the stronger mouth stretch movement) depending on facial orientation, recipient facial expression and play intensity, we calculated GLMM with binomial error distribution and logit link function. AU26 versus AU27 was imputed as the binary response factor. The identity of focal and recipient and the play bout number were added to the model as random factors. The model was slightly overdispersed (i.e., more variance than expected by the standard model), so we added an OMF-level random factor (1 | OMF), where OMF is a vector from 1 to the total number of observations (247) (Bolker, 2008)

The full model retaining all factors provided the best fit for the data (see Table 2). The full model had the lowest AIC (148.1) and removal of any of the factors resulted in a significant change to the model and when compared to the null model (facial orientation vs null model ANOVA F1 = 17.279, P < 0.001; recipient facial expression vs null model ANOVA: F1 = 138.01, P < 0.001; play intensity vs null model ANOVA: F1 = 16.374, P < 0.001). When comparing the full best model to the null model, the result was also highly significant (full model vs null model ANOVA F3 = 156.77, P < 0.001).

Within the fixed factors of the best model, facial orientation was the only significant factor (P < 0.05), being more associated with the stronger AU27 movement than AU26. However, when comparing models, recipient facial expression also had a strong (positive) significant influence (best model AIC: 148.1 vs model without facial orientation AIC: 157.5, ANOVA: F1 = 11.326, P < 0.001 and vs model without recipient facial expression AIC: 270.3, ANOVA: F1 = 124.16, P < 0.001), and play intensity had a weak negative influence (best model AIC: 148.1 vs model without play intensity AIC: 151.5, ANOVA: F1 = 5.352, P < 0.05) on the display of AU26 versus AU27. As low intensity was set as the baseline, a negative influence indicates that more intense play increased the likelihood of AU27 vs AU26. Therefore, playfaces were more likely to contain the more intense AU27 when the recipient was facing the sender (FTF), when the recipient produced an OMF, and when play was more intense (see Fig. 4).

Figure 4 Figure showing proportion of OMF with different intensities in different conditions.

Proportion of OMF (open mouth faces) with AU26 (jaw drop) and the more intense AU27 (mouth stretch) as (A) a function of facial orientation (FTF, face to face; Not FTF, not face to face), as (B) a function of the facial expression of the recipient (OMFR, recipient has OMF; nOMFR, recipient does not have OMF), and (C) as a function of play intensity (low and high).

Discussion

In the current study, orangutans were sensitive to the visual attention of their social partner when producing facial expressions. During social play, if the focal individual was facing their play partner with unobstructed visual access between the two individuals, open-mouth expressions (playfaces) were more intense and contained more component movements. Although previous studies have interpreted similar findings as evidence of intentional communication, we have a more cautious interpretation. These findings may not demonstrate intentionality beyond doubt, but do show that production of the playface is not an automated response to play (or simply a play behaviour itself) and instead is highly flexible and dependent on subtle characteristics of the social context.

Modified and increased production of visual signals as a function of the recipient’s attentional stance has been used as sufficient criteria for intentionality in previous studies of primate gesture (e.g., Hobaiter & Byrne, 2011a; Hobaiter & Byrne, 2011b), and as additional criteria in many other studies (e.g., Liebal, Call & Tomasello, 2004; Poss et al., 2006; Anderson, Kuwahata & Fujita, 2007; Genty & Byrne, 2010; Leavens, Russell & Hopkins, 2010). Sensitivity to attentional stance has been used as evidence for intentionality in signal production in this way, as it could demonstrate that the sender is intending to communicate (and is thus only communicating when the audience is receptive to the signal). If we accept this logic, and thus accept that this behavioural marker is a sufficient demonstration of intentional production, we must conclude that, like primate gestures, orangutan playfaces are produced intentionally.

It is possible, however, that sensitivity to attentional stance when producing signals demonstrates a degree of flexibility of production (which demonstrates it is not fully involuntary, reflexive or automatic) without the need to attribute goal-directed, purposeful communication. The decision to produce a playface may be voluntary in the same sense that the decision to bite, eat or run may be voluntary in some way—the animal is capable of inhibiting or increasing the behaviour if conditions suggest such an approach might be sensible. Although part of the continuum of intentionality, this is not the same as goal-directed, purposeful intention to communicate to another.

There are also other explanations that could be considered. First, face-to-face play could simply be more arousing and stimulate the production of more play (of which the playface is a component). Play intensity did have a weak positive impact on the intensity of the playface in our model, but also a negative impact on the complexity of the playface. Therefore, play intensity does seem to be one contributory factor (on intensity at least), but is not strong and does not influence the composition of the signal. Thus, increased playface production during face-to-face play is not simply a function of face-to-face play being more intense. Second, playfaces might be stronger during visual contact with the play partner due to reflexive mimicry of the partner’s facial expression (rapid facial mimicry: e.g., Davila Ross, Menzler & Zimmermann, 2008). The playfaces were more intense and more complex when the recipient was also producing a similar facial expression, so mimicry could play a part, but as facial orientation was also a strong factor in both models, mimicry cannot be the only explanation (although may play a role). Third, primates may be responsive to the face of their play partner during play as the face is a powerful stimulus for social primates and many species exhibit highly sophisticated facial processing skills (Parr, 2011). Although such abilities may be a stepping-stone towards (and necessary for) intentional communication, they might also be potential explanations for sensitivity to attentional stance in and of themselves. Primate individuals respond to the faces of others during social interaction on a regular basis, and so the face may act as a cue stimulating a response appropriate to the context. For example, primates respond to threat faces with submissive expressions or counter threats, and respond to subtle facial cues such as staring with overt behavioural responses (Yamagiwa, 1992). Such a response would not necessarily require intentionality.

Therefore, a number of alternative and additional explanations for sensitivity to visual attention in communication are plausible, and we should be cautious when concluding that complex intentional production has been demonstrated. Similar data, however, have been used to support the view that primates are capable of intentional production of gestures to achieve strategic social goals. The different research traditions underlying the study of facial expressions and gestures (Slocombe, Waller & Liebal, 2011; Liebal et al., 2014) may explain why different conclusions are being made in different fields. Primate gesture is often proposed as a potential precursor to human language (a debate which relies heavily on the data alluding to intentionality: Slocombe, Waller & Liebal, 2011). The vast majority of previous studies investigating intentionality in primate communication focus on gestures (as opposed to vocalisations or facial expressions): all of the 24 studies reviewed in Liebal et al. (2014) examined gesture, nine examined vocalisation, and seven examined facial expression. These data are consequently contributing to a body of work being cited as solid evidence that some species of primates can communicate in a flexible, goal-orientated, and intentional fashion, particularly with gestures. Furthermore, such data are being used as a crucial platform for the investigation of other language-like characteristics (e.g., Genty & Zuberbuhler, 2014; Hobaiter & Byrne, 2014). Thus, even if the gold standard is to use multiple criteria for the identification of intentionality, the implications of using only one or few of the most commonly applied criteria (which is the status quo) needs to be addressed.

Here, we demonstrated that production of orangutan facial expressions can be modified in response to the presence and visual attention of another. Such modification may not be evidence of intentional production, but it is nevertheless evidence of complexity within a communicative system that has been hitherto overlooked. The difference in intensity and complexity of the playface between facing and not facing conditions was only in degree, but extensive work on nonverbal behaviour in humans and other animals suggests that even rapid and subtle cues can have an impact on social interaction. Further research is important to determine whether this sensitivity to attentional stance does indeed have an important impact on consequent social interactions.

Facial expressions (in humans as well as other animals) have long been seen as rather fixed, biologically based expressions of emotion, reflecting the internal state of the sender, a theoretical stance reinforced since Darwin (1872). This may be true in some senses, but it is important that this assumption does not influence the generation and interpretation of data a priori. Also, in the absence of physiological data it may be just as difficult, and not necessarily more conservative, to conclude that a signal is emotional rather than intentional (Waller & Micheletta, 2013; Liebal et al., 2014) and so concluding that a signal is emotional may not necessarily be a more conservative explanation. Ultimately, a multimodal approach to primate communication might help overcome some of the constraints surrounding the study of primate communication by promoting behavioural criteria for the detection of intentionality to be used cautiously and consistently across species and communicative modalities. Importantly, sensitivity to attentional stance may not demonstrate intentionality akin to that used in human language, and so perhaps the significance of this trait in primate communication needs to be reconsidered. Yet claims of intentional communication in related primate species abound in language evolution debates, and thus caution in identifying intentionality is crucial.

Supplemental Information

Supplemental Information 1 Dataset

Click here for additional data file.

Recordings were obtained at Sepilok Orangutan Rehabilitation Centre and Kabili Sepilok Forest Reserve, following the approval for research by Sabah Wildlife Department and Economic Planning Unit, Malaysia. Thanks go to E Zimmermann for contributions to the field research, to E Bosi, H Bernard, and S Alsisto for logistic help and to M Wessels for assisting with the data collection. We thank Jerome Micheletta, Juliane Kaminski and Jamie Whitehouse for comments on the manuscript.

Additional Information and Declarations

Competing Interests

Author Contributions

Animal Ethics

The authors declare there are no competing interests.

Bridget M. Waller conceived and designed the experiments, analyzed the data, wrote the paper, reviewed drafts of the paper.

Cátia C. Caeiro conceived and designed the experiments, performed the experiments, analyzed the data, wrote the paper, prepared figures and/or tables, reviewed drafts of the paper.

Marina Davila-Ross conceived and designed the experiments, analyzed the data, contributed reagents/materials/analysis tools, wrote the paper, reviewed drafts of the paper.

The following information was supplied relating to ethical approvals (i.e., approving body and any reference numbers):

Research permission was provided by Sabah Wildlife Department and Economic Planning Unit, Malaysia.

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
