# Peer review of "Orangutans modify facial displays depending on recipient attention"

_PeerJ, doi:10.7717/peerj.827_

## Round 0.1 · original submission · Minor Revisions

Your paper has now been carefully reviewed by three experts who converge on the opinion that the topic and the results are of interest to the readership of PeerJ. However, the reviewers have concerns with the framing and interpretation of the study, suggesting that the manuscript needs to do a better job of situating the current study in the recent experimental literature. Whereas there may not be many empirical reports explicitly linking emotion to intentionality, experimental work on both topics should be discussed and integrated, as suggested by Reviewer 2. Reviewer 3 also suggests that the definition and practical relevance of the results should be made clearer. I am in agreement with the reviewers and would like to invite a revision of this article provided that you can follow the reviewers' thoughtful suggestions and demonstrate more caution in the interpretation of the results. The last sentence of your abstract, for example, is appropriate.

I have a few additional points of my own. I am not convinced that you are demonstrating for the first time that a primate facial expression can be modified by the attentional states of an observer. Surely there is evidence concerning threat expressions? Even when the sender modifies an expression as a function of attention of the recipient, it cannot be concluded that this is because the sender intends to communicate, as the sender may simply be responding to particular cues, such as visible and open eyes. I agree with the reviewers that the alternative interpretations are discussed reasonably in the discussion, but not framed cautiously enough in the intro. As reviewer 3 notes, it reads a bit like backpedaling.

In addition, work on non-primate animals is neglected in the current discussion. You might find it useful to examine evidence for use of play signals as "meta-communication" devices; for example, the play posture of the dog. Why was triadic play excluded from analysis? Reliability for FTF vs. Not FTF is not very good, especially considering intra-reliability was used (see also comments from Reviewer 2). Can you comment on how disagreements were resolved?

I do not have issues with the data analysis or its interpretation as you justify the approach well and it seems rigorous and appropriate. However, based on reviewer 3's comments you must clearly distinguish between a significant factor as it impacts the outcome, and the significance of the factor within the model tested.

As per Reviewer 3's comment, I would avoid discussing arguments of parsimony as I think you mean parsimony in terms of ease of verbal explanation, rather than simplicity in design and such arguments are thus often misleading.

Something is missing from line 158.
Figures, tables and references are out of order.

Reviewer 1 ·

Basic reporting

The manuscript was complete, well-written, and was appropriately "self contained" for submission to PeerJ. The introduction was both interesting and parsimonious. The discussion of the various controversies in analyzing intentionality were well-explained. The reviewer believes the introduction (and discussion of controversies) works well with the author's discussion section, where potential other explanations for the results are identified. A grammatical error noted is in line 158 "The GLMM analysis allowed us to random factors to control for the fact that…"

Experimental design

The reviewer believes the research design was sufficient to address the research question. However, it would be interesting to note play differences between the nursery group and the released group, if any occurred. Additionally, a demographic breakdown of each group (could we assume the nursery group was composed of younger individuals?) would have been helpful.

Validity of the findings

Although the reviewer is not an expert in GLMM, the logic of the analysis was sound, and the results appeared thorough. Appropriate controls, non-parametric tests, and follow up analyses were conducted to make the data robust. The discussion of the results, and possible alternative explanations was insightful.

·

Basic reporting

This starts off with a bit of a false dichotomy – facial expressions are not necessarily one or the other (intentional or tied to emotions). Similarly, just because a playface is intentionally modifiable, does not mean it is produced intentionally or that it is not tied to emotion. This point is mentioned later, but the setup already leads the reader to perceive an all-or-nothing dichotomy.

Most importantly, the background on intentionality and facial expressions is not well reviewed and must be strengthened for the final paper. The initial dichotomy is set up by using Darwin as the only reference for stating that facial expressions are tied to emotions. Only later is it mentioned that more contemporary researchers have looked at intentionality of facial expressions, but none of these studies are reviewed. I think the intro must spend more time on the current findings relating to facial expressions and intentionality.

The discussion of alternative explanations is good – if the intro were re-written a bit to come in line with the discussion, it would be preferable.

Experimental design

Analysis: I’m not very familiar with GLMM, so I cannot comment on the appropriateness of the analysis, however, the results are well written for the most part and the findings seem to support the interpretation that play faces are different when the orangutans are facing each other.

Line 184 –I’m really not understanding how testing for onset latencies of AUs ensures that longer OMFs do not account for higher number of AUs. Can you explain this a little more?

Validity of the findings

Overall, the findings seem solid, but I’m concerned that the proportion of the two facial expressions seems to differ strongly in response to attentional stance (see Fig 4). The pattern seems to suggest that A27 might be more under voluntary control than A26 and that difference may account for the findings. I think the authors should analyze the two facial expressions separately to properly interpret the possibility that only one of these facial expressions (A27) follows the pattern they discuss.

Table 1: there are many more unscorable recipient OMFs, almost 45% of them, than I expected – this should be specifically mentioned in your methods and/or results sections. Can your data still be reliable with that proportion of missing data for one of the predictor variables?

Table 2: Play intensity had a (tiny) positive estimate for number of AUs and a negative estimate for AU26 or 27 – what does this mean?

Additional comments

Smaller issues:
Line 13 – grammar – generally accepted as produced intentionally

Line 49 – so there have been studies that investigated facial expression as an intentional production? The authors will need to review those in the intro.

Line 67 – yes, but this statement seems to go against what you’ve said above.

Line 149 – so the reliability measure was only intra-observer reliability? You did not verify that a second coder would agree?

Line 158 – grammar – randomize?

Line 308 – these numbers are different than the numbers in the intro – are you talking about a different set of studies? Why weren’t the facial expression studies reviewed in the intro, especially if there are 7 of them? What were their findings?

Reviewer 3 ·

Basic reporting

In general, I feel that the writing could be substantially improved. Currently, the introduction and discussion sections seem underdeveloped and lack a strong coherent thesis. Intentionality is a rich interpretation and more references could be included throughout the introduction to bolster claims. In the discussion, the authors go quickly to the interpretation of intentionality, but then seem to ‘back-pedal’ through alternative explanations, such that it is difficult to discern what they truly believe the interpretation of the results should be. A stronger concluding paragraph would help strengthen the overarching thesis of the paper.

Experimental design

No comments

Validity of the findings

I wonder about the practical significance of the data, and the way that they have been interpreted. For instance, several times, a factor was found to not be significant in the model, but then, because its inclusion showed a significant improvement in the model (as opposed to if it was excluded), it is then described as having had a significant impact, and I find this conflict misleading. It can’t be both significant and not significant at once (For an example, see the description of the role of play intensity at Lines 209-214).

Moreover, when significance was reached statistically, I wonder about the practical significance of the results (particularly because effect sizes are not included). For instance, the authors found that there was a significant difference in duration between the duration of open-mouth faces when an individual was facing versus not facing a conspecific. However, the mean values for these two conditions are 1.64 s and 1.09 s, respectively. Even though these values may be statistically different, I wonder what the meaning of a half second difference is in real-time communication between orangutans.

Additional comments

Minor concerns:

First sentence of abstract, L. 13, and throughout: The language could be improved. For instance, there are several instances of the word “as” followed by a verb, and these need to be corrected. For instance in the first line of the abstract “...are widely accepted as underpinned”.

L. 17 A concrete example might help bolster this claim

L. 37 A matter of style, but the word “rather” weakens the writer’s position

L 39: The word “while” implies a temporal relationship, and could be changed to “whereas”

L 49, the numerals should be changed to written words

L56: If play faces are preserved across phylogeny, do you feel they are also intentional in other species?

L 79 and beyond: It would be helpful to include your definitions of expression “intensity” and “complexity” somewhere when discussing your hypothesis

Method:

Why were data coded only from play contexts? Would it be helpful to show that this face – if context specific – does not occur in non-play contexts?

L 103: were cameras stationary or held by humans? If held by humans, did this impact the behaviour of subjects?

L113: Did instances in which food was involved occur? If so, were these excluded? (e.g., it seems possible that a play bout could occur during feeding times, particularly for those semi-free-ranging when they came into a social situation, but food also necessitates an opening of the mouth

L124: I think it is important and well justified that the recipient’s facial expression was included in the coding, but it would strengthen the paper if this measure was explicitly discussed and justified in the introduction and/or method section (possibly as a more developed discussion on the possibility of imitation)

L 142: Was the coder blind to experimental hypotheses? Was the second coder that helped with reliability also certified and blind to hypotheses?

L 131: Why was the play face coded and not other facial expressions?

Statistical Analysis: There is no mention of the alpha level used to determine significance

Results:

Many of the details in the Results section would be more appropriately placed in the “Statistical analysis” section of the Method

L 177 A reference to the play face (Figure 2) might be useful earlier in the paper, either in the Introduction or Method section

L180 I wonder about the ecological validity or significance of an OMF lasting 0.08 in duration. It is wonderful that technology affords us the opportunity to capture such things, but I wonder about the impact (or lack thereof) of such a brief behaviour for two conspecifics, in real-time interaction

L182: Again, even if it is statistically significant (and with an N of 247, it seems less than surprising that significance could easily be reached), I wonder about the practical significance of a difference between 1.6 s and 1.1 s)

L194: Again, an inclusion of the definitions for “complexity” and “intensity” earlier in the paper would be helpful

I see that individual identity was included in the model as a random factor, but it would be nice to see some discussion somewhere about individual variation, both because there is commonly such wide variation between individual primates, and also because several individuals in the study showed opposite patterns to the overall trends that are discussed (For instance, Rosalinda and Kam Chong had many more OMFs when a conspecific was not face to face than when one was).

Discussion:

Again, because of the ambiguity of several statistical results, I feel that the authors should tone down their discussion and interpretation of their results. Several factors were not significant in the model (even if their inclusion provided an improved fit), but are still discussed as though they are significant.

Again, this section begins with strong statements (perhaps overly strong, in light of the mixed and weak patterns in results) about the intentionality of the behaviour, but then retreats and never re-states conclusions, such that it is difficult to know where the authors stand on the topic.

Lines 328-331 There are probably theorists (although I am not personally one) that would argue that an explanation stemming from emotion would be far from parsimonious in animals.

Acknowledgments:

Lines 338 and 342 are the same.

---

## Round 0.2 · accepted · Accept

You have been very responsive to the last round of reviews and I believe this paper will make a nice contribution to the literature.